# Equipping Farrowing Pens with Straw Improves Maternal Behavior and Physiology of Min-Pig Hybrid Sows

**DOI:** 10.3390/ani10010105

**Published:** 2020-01-08

**Authors:** Chao Wang, Qian Han, Runze Liu, Wenbo Ji, Yanju Bi, Pengfei Wen, Ran Yi, Peng Zhao, Jun Bao, Honggui Liu

**Affiliations:** College of Animal Science and Technology, Northeast Agricultural University, Harbin 150030, China; wqclly@163.com (C.W.); baozi337hanqian@163.com (Q.H.); liu171996@163.com (R.L.); 18845047210@163.com (W.J.); yanju_bi@163.com (Y.B.); wenpengfei321@163.com (P.W.); yiranneau0605@163.com (R.Y.); 15146009893@163.com (P.Z.)

**Keywords:** enriched environment, min-pig hybrid sow, maternal behavior, physiology

## Abstract

**Simple Summary:**

This study used different sow breeds combined with an enriched environment to relieve stress and to improve the welfare level of sows. Both groups of sows were in loose-housed pens. Sows living in a barren environment (BE) without straw were compared with sows in farrowing crates that received 3.5 kg of straw daily (enriched environment (EE)). Compared with BE sows, EE sows showed more nest-building behavior prior to farrowing, more nursing behavior, and less ventral recumbency behavior during the first three days postpartum. Furthermore, compared with BE sows, EE sows tended to have higher concentrations of serum oxytocin and prolactin, while the concentration of cortisol was lower, suggesting an increase in maternal behavior and a reduction of stress in this group. In addition, the concentration of oxytocin and both the frequency and duration of prepartum nest-building behavior were higher in Duroc × Min and Landrace × Min sows compared with Landrace × Yorkshire sows. However, the concentration of prolactin was significantly lower in both Duroc × Min and Landrace × Min sows than in Landrace × Yorkshire sows, indicating that hybrid sows of Min-pig inherited good maternal characteristics. Based on these results, straw enrichment improved Min-pig hybrid sow welfare during farrowing and lactation.

**Abstract:**

This study investigated the effects of two factors, enriched environment (EE) and different crossbreeds, on the maternal behavior and physiology of Min-pig hybrid sows. The analysis was performed on a total of 72 multiparous sows, including Duroc × Min pig (DM), Landrace × Min pig (LM), and Landrace × Yorkshire (LY) sows, using a total of 24 sows per cross. The sows were housed in two different farrowing pens, one with straw (EE) and one without straw (barren environment (BE)). The results showed that nest-building behavior, including the frequency, total duration, and bout duration, was significantly higher in EE sows than in BE sows (*p* < 0.01). The frequency and duration of prepartum nest-building behavior were higher in DM and LM sows than in LY sows (*p* < 0.0001). During the first three days postpartum, EE sows spent a shorter time in ventral recumbency compared with BE sows (*p* < 0.05). The oxytocin (*p* < 0.05) and prolactin (*p* < 0.01) concentrations of EE sows were significantly higher than in BE sows; however, the concentration of cortisol followed the opposite (*p* < 0.01). The concentration of oxytocin was significantly higher in DM and LM sows than in LY sows (*p* < 0.01). In conclusion, both EE increased the expression of hormones related to parental behaviors and prenatal nesting and nursing behavior of sows. Furthermore, an EE can also reduce stress in sows. Min-pig hybrids may inherit highly advantageous characteristics of maternal behavior of Min-pig sows.

## 1. Introduction

The death of preweaning piglets is a major source of economic loss for the pig industry, and also a significant welfare issue [1,2]. Maternal crushing has been reported as the main reason for piglet mortality [2,3]. To protect piglets from crushing, farrowing crates are used on commercial pig farms, which restrict a sow’s mobility [2,3]. However, there is no evidence from larger studies, which would support that crated sows have a lower piglet mortality rate than penned sows [4,5]. It has even been suggested that farrowing crates severely impair the welfare and health of sows [6]. Piglet survival rates correlate with maternal behavior, which varies between individual sows due to genetics and the environment [7,8,9]. Research focusing on the effects of housing on maternal behavior suggested that an enriched farrowing pen environment increases the possibility that sows freely make contact with their piglets, express exploratory behavior, and are able to avoid crushing their piglets [10]. A supply of nesting materials, which is provided prepartum in farrowing houses (e.g., sawdust and straw), can increase the duration and frequency of nest-building behavior prior to farrowing. This has been shown to be beneficial to parturition and the expression of maternal behavior in early-lactating sows [11,12,13]. Sows placed in loose farrowing environments showed a longer duration of postpartum lateral lying, which increases piglets’ access to the sows’ udders and thus piglet suckling and survival [14]. In contrast, it has been reported that a farrowing pen with straw exerts no effect on suckling duration and frequency [15,16]. Overall, controversy persists regarding the effects of enriched environments (EE) on sow maternal behavior.

The prenatal nest-building behavior of sows is related to the concentration of prolactin in the circulating blood [13]. In addition, endogenous prolactin (PRL) and oxytocin (OT) can activate the nursing behavior in lactating sows [17]. OT has also been reported to play an important role in the improvement of maternal behavior [18]. Therefore, it is important to explore the relationship between hormones and sow behavior under the influence of environment and breed type to improve the breeding of sows.

De Leeuw et al. showed that the cortisol (COR) concentration of sows in farrowing pens with straw (EE) was significantly lower compared with sows in crates [19]. A study by Jarvis demonstrated that the COR concentration of sows gradually increased from 48 h prepartum to farrowing and was significantly lower than that of restricted sows 6–4 h prepartum [20]. However, Jarvis et al. reported that the difference in the concentration of COR of Large White × Landrace sows between D5 prenatal and D29 postpartum (weaning at D28) was not significant [21]. The influence of the breed needs to be increased to further verify this result.

However, modern pig production breeding focuses on sow reproductive performance and ignores sow maternal ability [22]. Breeds differ in their maternal ability and selection toward better maternal behavior can increase the survival rate of piglets. As an indigenous pig breed in northeastern China, the Minzhu (Min pig) has favorable characteristics in both reproductive performance (14.35 piglets per birth) and maternal ability [23,24]. Therefore, to utilize the maternal advantages of the Min pig, crosses (Landrace × Min pigs (LM), Duroc × Min pigs (DM), and Landrace × Large White (LY)) were used in this study. Posture, postural change, nursing behavior, and the overall performance of the three different breeds were compared between barren and enriched farrowing conditions to investigate the effects of genetics and the environment on maternal behavior and physiology of sows. This investigation may help to reduce piglet mortality and improve the health and welfare of both sows and piglets.

## 2. Materials and Methods

### 2.1. Animals

Seventy-two multiparous sows were selected (24 sows each of DM, LM, and LY) with different sow parities ranging from three to five. Every hybrid group was randomly moved into an enriched farrowing pen (EE; n = 12) or a barren farrowing pen (BE; n = 12), the number of sows among each parity was equality. The litter size between both environment and breed had no significant differences. The housing layout did not differ between enriched and barren farrowing pens. The only difference was that enriched farrowing pens were equipped with straw and barren farrowing pens were not.

### 2.2. Housing and Management

All sows were kept in welfare farrowing pens. Figure 1 presents the field layout and design of the farrowing pens. Each farrowing pen measured 7.68 m^2^ in total; the inside of the pen was 4.80 m long and 1.60 m wide. The door of the farrowing pen was 0.80 m wide and 1.20 m high. Each farrowing pen was divided into two areas: the sow area, which consisted of the parturition and active region and measured 6.4 m^2^, and the piglet activity and rest region, which measured 1.28 m^2^. Both were separated by a board that restricted the sow’s ability to reach the piglet activity and rest region. The piglet activity and rest region were equipped with a heat lamp and food for the piglets. The sow area was divided into parturition and active region by a grass board. Each pen was equipped with a feeding trough and two drinking areas, supplying drinking water for both sows and piglets via nipple drinkers. The sow nipple drinker was installed 0.40 m above the ground, and the piglets’ drinker was installed 0.20 m above the ground. The ground consisted of concrete and was covered with straw (3.50 ± 0.25 kg). All of the straw was provided at 08:00 h daily. The angle between the sides of the parturition area and the active region was 18°. Supplemental ground heating was provided for sows and piglets in both the parturition area and the piglet activity and rest region and was controlled in each area.

One week prior to the expected parturition date, sows were moved from the gestation house to farrowing pens that had been thoroughly rinsed and disinfected beforehand. The sows were free to drink ad libitum and were fed three times per day (at 06:00 h, 10:00 h, and 17:30 h) in the farrowing pen. All sows received a complete feed containing 12.9 MJ of DE/kg, 17.0% crude protein, 3.40% crude fat, and 1.0% lysine. A feed level of 3 kg/d was provided for sows one week before the expected parturition date and was decreased by 0.5 kg/d until parturition. The sows received 0.5 kg/d of the complete feed 1 d after parturition, after which, the feed level was increased by 0.5 kg/d until free feeding. The health of the piglets was assessed, the number of piglets with diarrhea was recorded, and sick piglets were removed at 06:30 h daily. All pens received routine cleaning and were disinfected twice per week.

Natural ventilation and lighting were used, and both the temperature and relative humidity of the farrowing pens were measured daily (at 08:00 h, 14:00 h, and 20:00 h) with a hygrothermograph (Kestrel 4000 Pocket Weather Tracker; Kestrel, Santa Cruz, CA, USA). The average daily temperature and humidity in were 19.2 °C and 68.3% in April, 21.5 °C and 72.6% in May, 24.6 °C and 78.3% in June, 27.4 °C and 83.5% in July, 28.9 °C and 89.4% in August, 20.8 °C and 76.4% in September, and 18.6 °C and 71.2% in October.

### 2.3. Behavioral Observations

A Noldus Observer XT behavior observation system (Noldus Information Technology, Wageningen, The Netherlands) was used in the farrowing house. The nest-building behavior of the sows (n = 72) was videotaped continuously, starting 72 h before the expected parturition date and ending with the birth of the first piglet (BFP). The prepartum nest-building behavior of sows, such as rooting, pawing, or arranging of straw, was separately observed continuously starting from 18 h before parturition. The other sow behaviors were continuously recorded after the BFP. The 24 h period after the BFP was called day 1 (d1) postpartum, the subsequent 24 h period was called day 2 (d2), and the following 24 h period was called day 3 (d3). From the second to the fifth week after parturition, all behavior was recorded on video. Focal sampling was conducted from continuous recordings from 08:00 to 10:00 h and 13:00 to 15:00 h on the third and sixth day of each week.

Behavioral parameters and their definitions are listed in Table 1. State behaviors include posture. Event behavior includes nursing behavior; nest-building behavior; posture change.

### 2.4. Physiological Indicators

Blood samples of sows (n = 72) were collected on the day before farrowing, the farrowing day, and on the fourth day of the postpartum weeks. Blood was collected from the auricular vein before feeding (10:00). During each collection, 6 mL of blood was gathered, stored in an EDTA tube, placed into a centrifuge (TD4C desktop centrifuge, Changsha Yingtai instrument Co., Ltd., Changsha City, China) and was centrifuged for 10 min at 2000 r/min. After centrifugation, the serum of samples was separated, labeled, and stored at −20 °C. The samples were tested for OT, PRL, and COR using an ELISA kit (R & D, Shanghai Jinma Experimental equipment Co., Ltd., Shanghai City, China).

### 2.5. Statistical Analysis

Data were preliminarily analyzed using Microsoft Excel 2013. State behavior was recorded as a percentage of the total observations, and event behavior was recorded as the frequency of the occurrences over a predetermined time period. A GLM procedure using SAS statistical software (SAS, Ver 9.4) was employed to examine the differences in nest-building behaviors and physiological indicators among the three crossbreeds in different environments. The following statistical model was used:Y_ij_ = μ + E_i_ + B_j_ + E_i_ × B_j_ + e_ij_,(1)
where Y_ij_ represents the observed value of the nest-building behavior, μ represents a general mean, E_i_ represents the environmental effect, B_j_ represents the fixed effect of the breeds, E_i_ × B_j_ represents the interaction between the environment and the breeds, and e_ij_ represents the residual effect.

The model for the postpartum behavioral differences among the three crossbreeds and different environments and observations over different time periods are included in the following effects:Y_ijk_ = μ + E_i_ + B_j_ + T_k_ + e_ij_,(2)
where Y_ijk_ represents the observed value of the nest-building behavior, μ represents the general mean, E_i_ represents the environmental effect, B_j_ represents the fixed effect of the breeds, and T_k_ represents the effect of observation days (d1, d2, and d3) and observation weeks (w2, w3, w4, and w5) after parturition, and e_ij_ represents the residual effect. All statistical analyses were considered significant at *p* < 0.05.

## 3. Results

### 3.1. Nest-Building Behavior

The treatment effects on the nesting behavior of prepartum sows are presented in Table 2. EE sows had a longer nesting period and single duration of nest-building behavior (DNB) compared with BE sows. Furthermore, the nest-building frequency was higher in EE than in BE (*p* < 0.01). The frequency and DNB prepartum were higher in DM and LM sows than in LY sows (*p* < 0.01). No differences were found in single DNB between crossbreeds (*p* > 0.05).

The durations of the nest-building behavior of crossbred sows in the 3 d prepartum differed between EE and BE (Figure 2). The duration of nest-building behavior of LY sows remained at baseline (EE vs. BE −18 h to 12 h vs. −18 h to 8 h) and then began to increase at −13 h and −9 h in EE and BE, respectively. LY sows in both environments peaked at −5 h (Figure 2). The duration of nest-building behavior in DM sows began to increase at −18 h and −15 h and peaked at −10 h and −6 h in EE and BE, respectively (Figure 2). The duration of nest-building behavior in LM sows began to increase at −15 h and −13 h in EE and BE, respectively, and then peaked at −11 h vs. −7 h in EE and BE. EE peaked much earlier than BE, and the duration of nest-building behavior lasted much longer. The duration of nest building in DM sows and LM sows began to increase and peaked much earlier than in LY sows.

### 3.2. Postpartum Behavior

Lateral recumbency during the first three days postpartum followed a gradually decreasing tendency and differed between the observation days (*p* < 0.0001; Table 3). The ventral recumbency of sows on the second and third days postpartum was higher than on the first day postpartum (*p* < 0.0001). Sows in EE spent a shorter time in ventral recumbency compared with sows in BE (*p* < 0.05). Neither the types of environments nor the observation times showed any effect on sitting and standing (*p* > 0.05). LY sows spent more time sitting (*p* < 0.01) and less time standing (*p* < 0.0001) during the first three days postpartum compared with DM and LM sows. During the first three days, EE sows had a higher frequency of postural change from lateral recumbency to other postures compared with BE sows (*p* < 0.05). The changes among the frequency of postural changes from standing to ventral recumbency were significant between crossbreeds (DM > LM > LY; *p* < 0.0001).

Lateral recumbency during week 2 postpartum was more frequent than during week 5 postpartum and followed a decreased tendency from weeks 2 to 5 (Figure 3); however, there were no differences between both environments nor between the crossbreeds (*p* = 0.093, *p* = 0.4781; Table 4). EE exerted a significant effect on reducing the time sows spent in ventral recumbency (*p* < 0.0001; Table 4), and DM sows performed more ventral recumbency behavior than LM and LY sows from weeks 2 to 5 postpartum (*p* < 0.05). No differences were observed in the time spent sitting in EE vs BE or between the observation times from weeks 2 to 5 postpartum (*p* > 0.05). However, DM sows spent less time sitting than LM and LY sows (*p* = 0.0054). Additionally, no differences were found in the frequency of postural changes from lateral recumbency to other postures and from ventral to lateral recumbency between crossbreeds and EE vs BE for all observation times (weeks 2 to 5 postpartum) (*p* > 0.05). The frequency of sitting to ventral recumbency in DM sows was lower than in LM and LY sows during weeks 2 to 5 postpartum (*p* = 0.0002). DM and LM sows exhibited a higher frequency of postural changes from standing to ventral recumbency than LY sows (*p* < 0.0001).

Compared with BE sows, EE sows exhibited a shorter premassage duration (*p* < 0.01), longer postmassage duration (*p* < 0.01), and higher nursing terminated frequency (*p* < 0.05; Table 5). However, no differences were found in either the duration of nursing (*p* = 0.8861) or the nursing frequency (*p* = 0.9603) between EE and BE. The duration of nursing in LM sows was longer than in LY sows (*p* < 0.01); however, it was not different from DM sows (*p* > 0.05). LM sows had a shorter premassage duration, a longer postmassage duration, and a higher frequency of nursing compared with DM and LY sows (*p* < 0.01). No differences were found in the frequency of nursing terminated by sows between crossbreeds (*p* = 0.1644).

### 3.3. Physiological Indexes

The results of the physiological indexes of sows are shown in Table 6. The concentrations of OT and PRL in the blood of EE sows were significantly higher than those of BE sows (*p* < 0.05), and the concentration of COR was significantly lower than in the BE group (*p* < 0.01). The concentrations of PRL in LY sows were significantly higher than in DM and LM sows (*p* < 0.01). There was no significant difference in the COR concentration among different breeds (*p* > 0.05). The concentration of PRL in the blood of sows gradually decreased with increasing time, except for the fourth and fifth weeks postpartum, where significant differences were found between other time intervals (*p* < 0.01; Table 7). The concentration of COR also decreased gradually, except on the farrowing day and the second week postpartum, where significant differences were found between other time intervals (*p* < 0.01).

## 4. Discussion

### 4.1. Sow Nest-Building Behavior

Sows had a strong desire to perform nest-building behavior within 24 h prior to parturition, and the execution of this behavior is important for parturition, postpartum behavior of the sows, and survival of the piglets [28]. The provision of nesting material can enhance nest-building behavior and improve maternal behavior of prepartum sows during early lactation [13,27]. In the present study, sows of the EE group started their nest-building behavior earlier, it peaked earlier, and had a higher frequency, longer duration, and single duration compared with BE sows. This suggests that the provision of nesting material can enhance nest-building behavior and improve the maternal behavior of prepartum sows during early lactation. This is in accordance with the research results of Yun et al. (2014, 2015) and Thodberg et al. (2002) [13,27,29].

DM and LM sows performed beginning and peak nest-building behavior earlier and had a higher frequency and duration of nest-building behavior. This suggests that Min pig hybrids inherited successful maternal behavior from their Min pigs parents [23,24].

### 4.2. Sow Postpartum Behavior

Postpartum sows mainly performed lateral recumbency behavior [24], which not only formed an important part of maternal behavior but also exerted a considerable effect on the nursing of postpartum sows and the safety of newborn piglets [14]. During early lactation, the duration of postpartum lateral recumbency not only provided a warm and safe environment for the piglets but also allowed them to move closer to the mother’s udder [14], an action that actively promoted nursing in sows. Ringgenberg et al. [30] reported that EE with increased pen area and provision of sufficient space for the sows to freely perform lateral recumbency behavior can improve the duration of the posture. In this study, this effect of EE on the duration of lateral recumbency was not observed. One reason may be that adequate space was already provided since the sows already performed lateral recumbency behavior in both EE and BE pens in this study.

The ventral recumbency behavior of sows is a signal to refuse nursing [31] and is the main tool of the gradual process of weaning. In this study, EE sows spent less time performing ventral recumbency behavior than BE sows during the first three days of lactation. This indicated that EE sows had a strong inclination to perform exploratory behavior, which is motivated by novelty-seeking and appetitive foraging [3,4]. This, in turn, enhanced the time the sows spent engaged in standing or walking behavior. This result can be verified by another result of this study, which demonstrated that EE sows spent more time standing or walking than BE sows during weeks 2 to 5 postpartum. However, no differences were observed for other postural behaviors between EE and BE. With increasing lactation time from the second week postpartum, the time sows spent in ventral recumbency gradually increased. This is consistent with the result of Valros et al. [32] who also showed that sows performed nursing refusal behavior. The results of the present study suggested that the sows accomplished the gradual process of weaning by spending more time in ventral recumbency, rather than by postural changes (lateral recumbency to other postures). However, the postural change in postpartum sows from the lateral recumbency to other postures was also regarded as a sign of nursing refusal [33]. In addition, EE had a significant effect on the increase of the frequency of postural changes from lateral recumbency to other postures. This was due to an increase of attention focused on the piglets by the sows in EE [30].

Piglet crushing occurs mostly within the first three days postpartum, especially within the first 24 h postpartum [34,35]. Usually, sitting is an unavoidable stage during the postural change from lying to standing [36]. McGlone et al. [37] reported a positive correlation between the time sows spent sitting and the frequency of piglet crushing. In this study, EE exerted no effect on the time sows spent sitting during the first three days postpartum, which is not consistent with the observations of Ringgenberg et al. and Jarvis et al., who showed that postpartum EE sows spent less time in the sitting posture [30,38]. These results suggested that sitting was related to the likelihood of contact between sows and piglets. Drake et al. reported that if communication between sows and piglets is hindered, this results in an early conflict between them and increases the time sows spend in a sitting posture [39]. In this study, welfare farrowing pens were provided for the sows, and they could freely contact their piglets. This suggests that the time the postpartum sows spent in lateral recumbency and sitting was affected by the form of the farrowing pens [2]. In the present study, the time LY sows spent in postpartum sitting was longer than that of both LM and DM sows. This implies that the risk of piglet crushing by Min-pig crossbreeds was lower than that of LY sows when there was a postural change from sitting to lateral recumbency and from ventral to sitting recumbency [23]. However, this study found no incident of piglet crushing by sows during both postural changes. As a result, further studies are needed to verify whether these two postural changes increase the risk of piglet crushing. Bedding improves the physical comfort of the floor, and—unless temperatures are high—straw enables pigs to somewhat control their microclimate thereby increasing thermal comfort [40]. Perhaps this is one of the reasons why fewer piglets were crushed.

The postural changes of postpartum sows, including ventral to lateral recumbency [41], sitting to lateral recumbency [25] and standing to lying [42], increase the risk of piglet crushing. This study found no effect of EE on the frequency of most postural change both during the first three days of lactation and from weeks 2 to 5 postpartum. However, Herskin et al. reported that the frequency of postural change could be reduced by the provision of sand or straw for farrowing pens [12]. The different results might be because this study provided straw before farrowing and Herskin et al. only provided straw during the lactation period. This was beneficial to the performance of sow nest-building behavior and affected the behavior of postpartum sows [12,13,27]. Another reason for the difference could be that this study used a different thickness and amount of straw. The sows’ postural change decreased when the sows lay in a comfortable environment. Additionally, the frequency of the postural change from standing to ventral recumbency was higher in LM and DM sows than in LY sows, which might be related to individual sow behavior. During farrowing, the sows stood up to express carefulness for the newborn piglet via smell and hogging when each piglet was born [3,23]. In the present study, the frequency of this behavior was much higher in LM and DM sows than in LY sows, which resulted in an increase in the frequency of the postural change from standing to ventral recumbency.

The nursing behavior of sows was important and was the sole source of nutrition for piglet survival. The frequency and duration of nursing significantly impacted the intake of milk by piglets [43]. In the present study, EE exerted no effect on the frequency and duration of nursing during weeks 2 to 5 postpartum, which contrasts with the results of Herskin et al. and Cronin et al. [12,17]. They reported that the frequency and duration of nursing could be increased by providing sand or straw in the farrowing pens. The different results in the frequency and duration of nursing could be explained by the size of the farrowing pens, and we suspect that the provision of straw had only a small effect on the nursing behavior of sows. Additionally, the frequency and duration of nursing in LM sows were higher than in LY sows, which suggests that sow maternal behavior was affected by genetic factors. In this study, the duration of premassage was shorter in EE compared with BE, while the duration of postmassage was longer in EE than in BE. The difference in the duration of massage between both environments could be explained by the EE sows increasing the frequency of nursing termination due to postural changes [5,6,7]. This was confirmed by the present study, which showed that the frequency of nursing termination by sows was higher in EE than in BE. LM sows had a shorter premassage duration compared with DM and LY sows; however, the duration of postmassage showed the opposite result. No difference was observed in the frequency between the three crossbreeds. This suggests that LM sows inherited the maternal behavior of Min pigs [24]. With increasing lactation time, the duration and frequency of nursing gradually decreased, especially where significant differences in the duration of nursing existed between each week, which agrees with previous studies [44,45]. This might suggest that sows gradually achieved the process of weaning [24].

### 4.3. Sow Physiology

In the present study, the concentrations of OT and PRL in the blood of sows were related to prenatal nest-building behavior [13,27]. The concentrations of OT and PRL in EE sows were significantly higher than those in BE sows, which is consistent with the results of Yun et al. [13]. This may be due to the sufficient nesting material, which was provided in the EE treatment, which thus enabled the sows to fully express their natural nest-building behavior. This was also confirmed by the results of nest-building behavior tests, which suggest that this behavior helps to increase the concentration of OT in the blood [13]. The higher concentration of OT in DM and LM sows, as opposed to LY sows, may be the cause of their higher engagement in prenatal and nest-building behaviors. Studies have shown a positive correlation between PRL and the OT concentration in the blood of prenatal sows [27]. In addition, the concentration of PRL gradually decreased with increasing lactation time, which is similar to the results of Liu [24]. This may be because the release of PRL is mainly triggered through sow nipple massage [46] and usually reaches its maximum level 10–20 min after the start of nursing [47]. However, with increasing lactation time, the sows’ nipple massage was prevented by a change in posture and the concentration of PRL decreased gradually, which was confirmed by the results of the lactation test. The concentration of PRL in the blood of DM and LM sows was significantly lower than in LY sows. The results of physiology data combined with production performance and sow behaviors showed that the crossbred sows showed no advantage in maternal behavior [48,49].

The concentration of COR in EE sows was significantly lower than in BE sows, which is consistent with the results of De Leeuw et al. [19]. This may be because the restricted nest-building behavior in BE sows, physiological stress, and increased activity of the HPA axis, increased the concentration of COR. Thus, the concentration of COR increased [50,51]. Furthermore, the concentration of COR decreased with increasing lactation time, which may be due to the adaptive change of the HPA axis during continuous stress. The level of COR in the blood gradually returned to normal, prestress levels.

## 5. Conclusions

In conclusion, an EE may not only induce the expression of physiological hormones related to parental behaviors but may also stimulate the prenatal nesting behavior of sows, which is beneficial for the expression of nursing behavior. Furthermore, EE can also reduce stress in sows. Compared with LY sows, DM and LM sows showed more advantageous maternal behavior characteristics. As a result, EE was beneficial for Min-pig hybrid sows in performing several maternal behaviors, which might exert an important influence on the survival of piglets before weaning.

## Figures and Tables

**Figure 1 animals-10-00105-f001:**
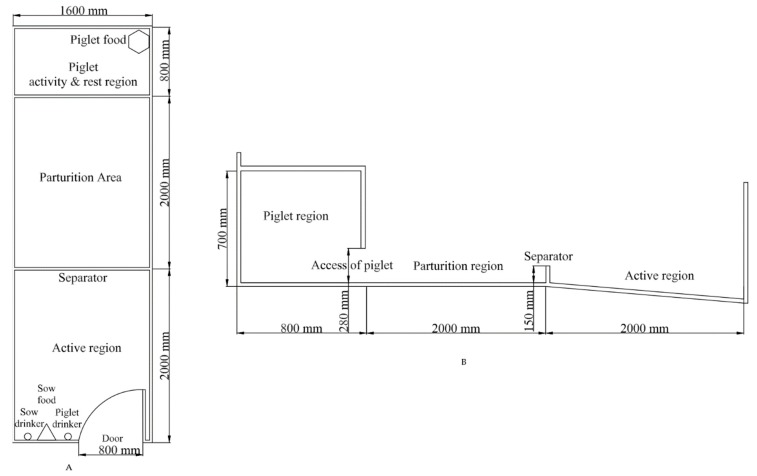
Planar structure chart of the welfare farrowing pen. (**A**) vertical view of the farrowing pen and (**B**) dimensions of the fallowing pen.

**Figure 2 animals-10-00105-f002:**
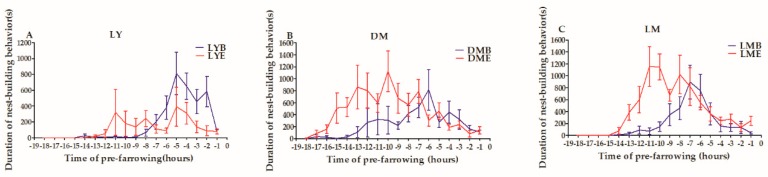
Trends of nesting behavior of LY sows (**A**), DM sows (**B**), and LM sows (**C**) during the first three days of litter bearing in EE and BE (LY = Landrace × Yorkshire sows, DM = Duroc × Min sows, LM = Landrace × Min sows, EE = Enriched environment, BE = Barren environment).

**Figure 3 animals-10-00105-f003:**
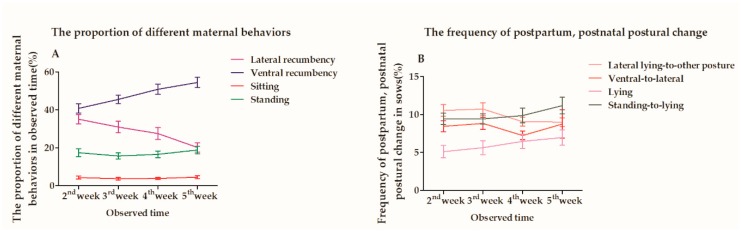
(**A**) Proportion of different maternal behaviors during the observed time from the second week to the fifth week postpartum in sows. (**B**) Frequency of postpartum, postnatal postural change in sows.

**Table 1 animals-10-00105-t001:** Behavioral parameters and their definitions.

Behavioral Parameters	Definitions
**Nest-building behavior [13]**	
Rooting	Pushing the floor or attempting to turn up the ground with the snout
Pawing	Attempting to scrape the floor with the front legs
Arranging	Manipulating or arranging nesting materials with the snout or mouth
Duration of nest-building	Actively performing nest-building for longer than 5 s between not performing nest-building for longer than 30 s
Nest-building frequency	Number of times when sows perform nest-building in a predetermined time period [16]
**Posture [23,25,26]**	
Standing or walking	Body weight supported by four legs; motionless or walking; including walking combined with other behaviors
Ventral recumbency	Sow’s chest and abdomen touching the floor and front legs stretched or folded beneath the body
Lateral recumbency	Sow’s head, ear, scapular, and waist all touching the floor and all four legs visible
Sitting	Most of the sow’s body weight and the posterior of her body trunk were in contact with and supported by the ground
**Posture change [23,25]**	
Lateral lying-to-other posture	Posture change from lateral lying to ventral lying, sitting, or standing
Ventral-to-lateral lying	Posture change from ventral lying to lateral lying
Sitting-to-lying	Posture change from sitting to lying, including ventral lying and lateral lying
Standing-to-lying	Posture change from standing to lying, including ventral lying and lateral lying
**Nursing behavior [23,27]**	
Duration of nursing	Duration was identified by the rapid suckling of piglets with milk ejection, calculated in seconds
Nursing frequency	Number of times when sows released milk within a predetermined time period
Duration of premassage	Premassage was considered to start when at least half the litter started to massage the udder and ended once the milk was ejected
Duration of postmassage	Postmassage was initiated when the milk ejection ended and terminated when at least half the litter stopped massaging the udder. If the sow changed postures, such as by rolling over or by walking away from the piglets, the postmassage was finished
Nursing terminated by the sow	After milk ejection, piglet suckling was terminated by a posture change of the sow

**Table 2 animals-10-00105-t002:** Nest-building behavior of prepartum sows.

Treatment		Behavior		
		Frequency (n = 12)	Duration (min) (n = 12)	Single Duration (s) (n = 12)
LYB		38.8 ± 16.46 ^Aa^	45.7 ± 33.57 ^Aa^	72.3 ± 98.17 ^A^
LYE		45.8 ± 28.65 ^Ba^	55.4 ± 29.12 ^Ba^	85.9 ± 116.19 ^B^
DMB		62.3 ± 22.18 ^Ab^	80.9 ± 48.18 ^Ab^	80.8 ± 107.93 ^A^
DME		98.3 ± 8.69 ^Bb^	158.9 ± 33.48 ^Bb^	97.6 ± 139.35 ^B^
LMB		65.7 ± 9.07 ^Ab^	92.1 ± 31.14 ^Ab^	82.5 ± 92.41 ^A^
LME		78.8 ± 21.22 ^Bb^	144.8 ± 27.52 ^Bb^	100.9 ± 133.64 ^B^
*p*-Value	E	0.0063	0.0003	0.0006
	B	<0.0001	<0.0001	0.23

^A,B,a,b,c^*p* < 0.05 (LYB = Landrace × Yorkshire sows in the Barren environment, LYE = Landrace × Yorkshire sows in the Enriched environment, DMB = Duroc × Min sows in the Barren environment, DME = Duroc × Min sows in the Enriched environment, LMB = Landrace × Min sows in the Barren environment, LME = Landrace × Min sows in the Enriched environment, E = environmental effect, B = fixed effect of the breeds). Data are represented as means ± SD.

**Table 3 animals-10-00105-t003:** Maternal behavior of sows during the first three days of lactation.

Behavior	Environment		Crossbreed			Day			*p*-Value		
	EE (n = 12)	BE (n = 12)	DM (n = 12)	LM (n = 12)	LY (n = 12)	d1 (n = 12)	d2 (n = 12)	d3 (n = 12)	E	C	D
Lateral Recumbency (%)	76.9 ± 11.16	73.3 ± 11.98	74.2 ± 11.41	74.2 ± 11.87	77.0 ± 11.85	82.1 ± 7.26 ^a^	74.4 ± 8.75 ^b^	68.5 ± 14.05 ^c^	0.08 F_0.05_(1,70) = 2.75	0.35 F_0.05_(2,69) = 1.04	<0.0001 F_0.05_(2,69) = 17.46
Ventral Recumbency (%)	14.6 ± 8.63 ^a^	17.5 ± 7.83 ^b^	17.9 ± 10.03	15.3 ± 7.49	14.9 ± 6.96	11.1 ± 5.46 ^a^	17.3 ± 7.43 ^b^	19.9 ± 9.33 ^b^	0.05 F_0.05_(1,70) = 3.91	0.16 F_0.05_(2,69) = 1.78	<0.0001 F_0.05_(2,69) = 11.39
Sitting (%)	1.0 ±0.93	1.2 ± 1.1	0.9 ± 0.71 ^a^	0.8 ± 0.97 ^a^	1.7 ± 1.16 ^b^	0.9 ± 0.80	1.1 ± 1.17	1.4 ± 1.04	0.40 F_0.05_(1,70) = 0.67	<0.05 F_0.05_(2,69) = 3.72	0.16 F_0.05_(2,69) = 1.79
Standing (%)	5.9 ± 2.45	6.7 ± 3.18	6.5 ± 2.58 ^a^	7.6 ± 3.23 ^a^	4.7 ± 1.84 ^b^	5.5 ± 2.75	6.5 ± 2.49	6.7 ± 3.24	0.1148 F_0.05_(1,70) = 2.21	<0.0001 F_0.05_(2,69) = 8.94	0.09 F_0.05_(2,69) = 2.74
Lateral recumbency to Others	19.0 ± 7.99 ^a^	15.2 ± 7.15 ^b^	19.2 ± 6.56	17.1 ± 8.31	14.9 ± 7.99	17.9 ± 8.18	15.1 ± 7.44	18.3 ± 7.51	0.01 F_0.05_(1,70) = 6.07	0.08 F_0.05_(2,69) = 2.81	0.16 F_0.05_(2,69) = 1.74
Standing to Lateral recumbency	15.7 ± 7.63	13.8 ± 6.22	16.2 ± 6.93	14.3 ± 7.13	13.6 ± 6.81	14.2 ± 7.16	14.2 ± 6.63	15.9 ± 7.22	0.18 F_0.05_(1,70) = 1.66	0.31 F_0.05_(2,69) = 1.18	0.50 F_0.05_(2,69) = 0.48
Sitting to Ventral recumbency	6.8 ± 7.07	4.8 ± 4.98	5.3 ± 5.68	5.0 ± 5.6	7.2 ± 7.12	5.7 ± 5.65	6.4 ± 7.62	5.3 ± 4.98	0.09 F_0.05_(1,70) = 2.84	0.26 F_0.05_(2,69) = 1.33	0.72 F_0.05_(2,69) = 0.39
Standing to Ventral recumbency	14.3 ± 5.56	15.1 ± 7.28	18.5 ± 4.95 ^a^	15.2 ± 7.43 ^b^	10.2 ± 3.36 ^c^	16.3 ± 7.61	13.3 ± 5.98	14.5 ± 5.36	0.41 F_0.05_(1,70) = 0.64	<0.0001 F_0.05_(2,69) = 11.90	0.07 F_0.05_(2,69) = 2.71

^a,b,c^*p* < 0.05 Abbreviations: EE = sows in the Enriched Environment, BE = sows in the Barren Environment, DM = Duroc × Min sows, LM = Landrace × Min sows, LY = Landrace × Yorkshire sows, d1 = first day postpartum, d2 = second day postpartum, d3 = thirdly day postpartum, E = *p*-value of the environment, C = *p*-value of crossbreed, D = *p*-value of day. Data are represented as means ± SD.

**Table 4 animals-10-00105-t004:** Maternal behavior of sows from weeks 2 to 5 postpartum.

Behavior	Environment		Crossbreed			*p*-Value	
	EE (n = 12)	BE (n = 12)	DM (n = 12)	LM (n = 12)	LY (n = 12)	E	C
Lateral Recumbency (%)	32.1 ± 15.94	27.5 ± 17.55	28.4 ± 16.85	32.1 ± 15.81	28.9 ± 17.97	0.09 F_0.05_(1,70) = 2.44	0.48 F_0.05_(2,69) = 0.65
Ventral Recumbency (%)	41.7 ± 13.75 ^a^	52.2 ± 14.62 ^b^	50.9 ± 15.82 ^a^	43.5 ± 14.95 ^b^	46.4 ± 13.85 ^ab^	<0.0001 F_0.05_(1,70) = 15.3	0.03 F_0.05_(2,69) = 4.15
Sitting (%)	4.5 ± 4.66	3.9 ± 4.06	2.5 ± 2.90 ^a^	4.7 ± 5.07 ^b^	5.2 ± 4.43 ^b^	0.40 F_0.05_(1,70) = 0.67	0.0054 F_0.05_(2,69) = 5.39
Standing (%)	18.0 ± 11.49 ^a^	14.7 ± 6.66 ^b^	15.9 ± 8.83	17.4 ± 11.53	15.7 ± 7.87	0.04 F_0.05_(1,70) = 4.15	0.56 F_0.05_(2,69) = 0.57
Lateral recumbency to Others	10.3 ± 3.91	10.8 ± 3.79	11.5 ± 4.00	10.3 ± 4.04	9.9 ± 3.36	0.43 F_0.05_(1,70) = 0.86	0.09 F_0.05_(2,69) = 2.44
Standing to Lateral recumbency	8.1 ± 4.33	9.6 ± 8.17	8.4 ± 4.71	9.9 ± 9.92	8.3 ± 2.96	0.17 F_0.05_(1,70) = 1.81	0.41 F_0.05_(2,69) = 0.68
Sitting to Ventral recumbency	6.6 ± 6.23	6.6 ± 4.59	4.2 ± 3.89 ^a^	8.7 ± 5.74 ^b^	6.9 ± 5.62 ^b^	0.97 F_0.05_(1,70) = 0.11	0.0002 F_0.05_(2,69) = 8.53
Standing to Ventral recumbency	10.0 ± 4.52 ^a^	11.5 ± 5.51 ^b^	12.0 ± 5.42 ^a^	12.5 ± 4.66 ^a^	7.8 ± 3.68 ^b^	0.04 F_0.05_(1,70) = 4.15	<0.0001 F_0.05_(2,69) = 11.60

^a,b,c^*p* < 0.05. Abbreviations: EE = sows in the Enriched Environment, BE = sows in the Barren Environment, DM = Duroc × Min sows, LM = Landrace × Min sows, LY = Landrace × Yorkshire sows, E = *p*-value of the Environment, C = *p*-value of Crossbreed, D = *p*-value of Day. Data are represented as means ± SD.

**Table 5 animals-10-00105-t005:** Nursing behavior of sows from weeks 2 to 5 postpartum.

Behavior	Environment		Crossbreed			*p*-Value	
	EE (n = 12)	BE (n = 12)	DM (n = 12)	LM (n = 12)	LY (n = 12)	E (n = 12)	C (n = 12)
Duration of nursing (s)	14.9 ± 3.65	14.9 ± 3.50	14.9 ± 3.56 ^ab^	15.4 ± 3.62 ^a^	14.5 ± 3.48 ^b^	0.89 F_0.05_(1,70) = 0.04	0.0005 F_0.05_(2,69) = 8.13
Duration of premassage (min)	1.6 ± 0.53 ^a^	1.8 ± 0.59 ^b^	1.8 ± 0.57 ^a^	1.6 ± 0.53 ^b^	1.7 ± 0.59 ^a^	<0.0001 F_0.05_(1,70) = 16.1	<0.0001 F_0.05_(2,69) = 15.9
Duration of postmassage (min)	2.7 ± 2.65 ^a^	2.2 ± 2.41 ^b^	2.2 ± 2.23 ^b^	2.8 ± 2.83 ^a^	2.3 ± 2.5 ^b^	0.0002 F_0.05_(1,70) = 8.53	0.0053 F_0.05_(2,69) = 7.93
Frequency of nursing	8.7 ± 1.70	8.7 ± 1.94	8.5 ± 1.49 ^b^	9.5 ± 1.82 ^a^	8.0 ± 1.85 ^b^	0.96 F_0.05_(1,70) = 0.08	0.0001 F_0.05_(2,69) = 15.3
Frequency of nursing terminated	7.1 ± 2.25 ^a^	6.4 ± 1.73 ^b^	67.0 ± 1.73	7.0 ± 2.27	6.3 ± 2.01	0.04 F_0.05_(1,70) = 4.09	0.16 F_0.05_(2,69) = 1.79

^a,b,c^*p* < 0.05. Abbreviations: EE = sows in the Enriched Environment, BE = sows in the Barren Environment, DM = Duroc × Min sows, LM = Landrace × Min sows, LY = Landrace × Yorkshire sows, E = *p*-value of the Environment, C = *p*-value of Crossbreed, D = *p*-value of Day. Data are represented as means ± SD.

**Table 6 animals-10-00105-t006:** Stress and immune levels of sows for different environments and crossbreeds.

Physiological Indexes	Environment		Crossbreeds			*p*-Value	
	EE (n = 12)	BE (n = 12)	DM (n = 12)	LM (n = 12)	LY (n = 12)	E	C
OT (ng/mL)	44.3 ± 5.67 ^a^	40.8 ± 5.02 ^b^	44.4 ± 5.30 ^a^	44.5 ± 5.30 ^a^	38.7 ± 4.31 ^b^	0.03 F_0.05_(1,70) = 5.12	0.0066 F_0.05_(2,69) = 7.43
PRL (ng/mL)	19.6 ± 3.61 ^a^	17.3 ± 3.50 ^b^	18.2 ± 3.50 ^a^	17.4 ± 3.49 ^b^	19.8 ± 3.83 ^c^	<0.0001 F_0.05_(1,70) = 57.00	<0.0001 F_0.05_(2,69) = 41.09
COR (ng/mL)	90.5 ± 16.74 ^a^	101.5 ± 15.63 ^b^	97.4 ± 16.78	94.5 ± 15.66	96.0 ± 18.75	<0.0001 F_0.05_(1,70) = 69.31	0.17 F_0.05_(2,69) = 4.94

^a,b,c^*p* < 0.05. Abbreviations: EE = sows in the Enriched Environment, BE = sows in the Barren Environment, DM = Duroc × Min sows, LM = Landrace × Min sows, LY = Landrace × Yorkshire sows, E = *p*-value of the Environment, C = *p*-value of Crossbreed, OT = concentration of OxyTocin, PRL = concentration of PRoLactin, COR = concentration of CORtisol. Data are represented as means ± SD.

**Table 7 animals-10-00105-t007:** Physiological statuses and immune levels of sows at different stages.

Physiological Indexes	Day before Farrowing (n = 12)	Farrowing Day (n = 12)	2nd Week Postpartum (n = 12)	3rd Week Postpartum (n = 12)	4th Week Postpartum (n = 12)	5th Week Postpartum (n = 12)	*p*-Value
PRL (ng/mL)	23.2 ± 2.91 ^a^	20.9 ± 2.16 ^b^	19.2 ± 2.52 ^c^	17.0 ± 2.10 ^d^	15.3 ± 1.89 ^e^	15.1 ± 1.84 ^e^	<0.0001 F_0.05_(5,66) = 27.49
COR (ng/mL)	116.2 ± 11.58 ^a^	105.7 ± 9.62 ^b^	100.9 ± 11.42 ^b^	92.6 ± 12.83 ^c^	83.9 ± 8.96 ^d^	76.6 ± 9.84 ^e^	<0.0001 F_0.05_(5,66) = 38.93

^a,b^*p* < 0.05. PRL = concentration of PRoLactin, COR = concentration of CORtisol. Data are represented as means ± SD.

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
