# Peer review of "Equipping Farrowing Pens with Straw Improves Maternal Behavior and Physiology of Min-Pig Hybrid Sows"

_animals, 2020, doi:10.3390/ani10010105_

Round 1

Reviewer 1 Report

Congratulations to the authors on this article. The topic of farrowing systems - barren versus enrichment - is a very important and topical one in many countries. 

The article is written in a very clear way, both the methodology, as the results with the statistical analysis are clearly described. The conclusions are supported by the results and in general terms similar to previous studies. Where they differ, the authors clearly indicate this and give possible explanations. 

The only part I missed in the article was some information on the health and welfare of the piglets - mortality/health problems seen/ growth rates/etc. Maybe this is kept for a separate article? It would be good to just include at least one sentence on whether or not the EE versus the BE had any impact on piglet health, welfare and productivity. 

But once again, congratulations for this excellent article.  

Author Response

Reviewer comments:

The only part I missed in the article was some information on the health and welfare of the piglets - mortality/health problems seen/ growth rates/etc. Maybe this is kept for a separate article? It would be good to just include at least one sentence on whether or not the EE versus the BE had any impact on piglet health, welfare and productivity. 

Response: Thank you for your advice. The topic and purpose of this paper is to study the behavior and physiology of sows, hence I didn’t mention the information on the health and welfare of the piglets. The date of piglets as following. And I had added the sentence to the manuscripts (line 96): The litter size between both environment and breed had no significant differences.

LYB

LYE

DMB

DME

LMB

LME

E

B

D

Litter size

13.33±2.73

13.33±1.86

14.00±2.19

13.33±1.86

15.33±2.50

13.50±2.42

0.2829

0.5013

0.615

Reviewer 2 Report

Equipping farrowing pens with straw improves 2 maternal behavior and physiology of Min-pig hybrid 3 sows

General comments

This manuscript looks at the effect of enrichment (nesting material) and genetics (three different crossbreeds) on the behaviour and physiology of commercially farmed pigs. This work is important and adds to the current literature in this area, but in my opinion, needs re-analysis to draw the conclusions the authors currently state. Analysing nesting as an aggregation of all nesting behaviours means that significant results are bound to be seen, as enriched pigs have the opportunity to build nests and barren pigs do not. A stronger and more useful analysis would be to separate out manipulation of nesting behaviours (and analyse the effect of breed on these) from non-manipulation behaviours (rooting and pawing) which can be analysed for breed and enrichment effects. Parity is currently not included in the model, but could be an important factor of maternal behaviour. Time should be included in both models if the authors wish to explore effects of time on pre-partum behaviour (lines 183-191). Throughout the results, the interactions are not reported, though they are included in the methods. Please could the authors make it clear whether interactions were significant and if not, whether the interaction was removed before a model re-run? I would suggest doing this with the re-analysis. In addition, effect measures are not reported with only p-values throughout the text.

Throughout the paper uses abbreviations which are confusing and make the manuscript difficult to follow. I suggest replacing these abbreviations with the full terms for the physiological measures and behaviours (crossbreed abbreviations could be kept).

I would like to see this paper published as the study is comprehensive and important for refinement and sow welfare, but the manuscript needs some major revision. I hope that my comments can help.

My detailed comments are as follows.

Detailed comments

Abstract

Add in background detail about why enrichment is used for farrowing sows.

Landrace x Yorkshire is incorrectly abbreviated

Lines 41/42: I don’t think this claim can be substantiated by the results of the study. Inheritance of traits was not studied.

Introduction

50: Does this refer to piglet or sow mortality?

51: Remove “thus”

75/76: This sentence is not clear. Breed is not mentioned when citing the previous two studies, did they study breed?

80: Consider rewording “outstanding”

130 – 140: It would be helpful here to detail the total number of hours of footage which were observed and the sampling schedule. Was footage watched continuously or in sampling intervals? From summing the hours stated, it seems that 3600 hours of footage were watched, is this correct?

142: How was it known that the sow would farrow the next day? Was blood taken every day and retrospectively analysed when farrowing date was know?

150: Which behaviours were state and which were events. Maybe this could be defined and added to the ethogram.

168 (ethogram):

Nest building duration – where does the time frame of 5 and 30s come from?  What happened if the sow spent longer than 30s building a nest?

Nest building frequency – please add the pre-determined time period in here

Suggestion (personal preference): Consider left aligning the table to make it easier to read.

Results

Table 2: Single duration of nesting is not defined in the methods, please add this in to make the results easier to understand.

Figure 2: Please add the unit of measurement on the y axes.

195-211: This section feels repetitive since all of the results are reported in the table. Perhaps instead the authors could summarise the key findings here and direct the reader to the table for full results.

Table 3: Were percentages/proportions of behaviours used in the model? Percentages are bounded by 0 and 100 and as such are not suitable for the type of model used. Beta regressions may be a useful model or poisson count models for frequencies.

222: I think this p-values may be incorrect. I cannot see it in the table and it does not seem to match the result stated.

224: Where does this p-value come from?

Discussion

A re-analysis of results may well change the outcome of the discussion. I will be happy to provide a review of the discussion after the re-analysis of the data.

Author Response

Reviewer: 2

Open Review

English language and style

( ) Extensive editing of English language and style required
( ) Moderate English changes required
(x) English language and style are fine/minor spell check required
( ) I don't feel qualified to judge about the English language and style

Yes

Can be improved

Must be improved

Not applicable

Does the introduction provide sufficient background and include all relevant references?

( )

(x)

( )

( )

Is the research design appropriate?

( )

(x)

( )

( )

Are the methods adequately described?

( )

( )

(x)

( )

Are the results clearly presented?

( )

( )

(x)

( )

Are the conclusions supported by the results?

( )

( )

( )

(x)

Comments and Suggestions for Authors

Equipping farrowing pens with straw improves 2 maternal behavior and physiology of Min-pig hybrid 3 sows

General comments

This manuscript looks at the effect of enrichment (nesting material) and genetics (three different crossbreeds) on the behaviour and physiology of commercially farmed pigs. This work is important and adds to the current literature in this area, but in my opinion, needs re-analysis to draw the conclusions the authors currently state. Analysing nesting as an aggregation of all nesting behaviours means that significant results are bound to be seen, as enriched pigs have the opportunity to build nests and barren pigs do not. A stronger and more useful analysis would be to separate out manipulation of nesting behaviours (and analyse the effect of breed on these) from non-manipulation behaviours (rooting and pawing) which can be analysed for breed and enrichment effects. Parity is currently not included in the model, but could be an important factor of maternal behaviour. Time should be included in both models if the authors wish to explore effects of time on pre-partum behaviour (lines 183-191). Throughout the results, the interactions are not reported, though they are included in th e methods. Please could the authors make it clear whether interactions were significant and if not, whether the interaction was removed before a model re-run? I would suggest doing this with the re-analysis. In addition, effect measures are not reported with only p-values throughout the text.

Throughout the paper uses abbreviations which are confusing and make the manuscript difficult to follow. I suggest replacing these abbreviations with the full terms for the physiological measures and behaviours (crossbreed abbreviations could be kept).

I would like to see this paper published as the study is comprehensive and important for refinement and sow welfare, but the manuscript needs some major revision. I hope that my comments can help.

My detailed comments are as follows.

Detailed comments

Abstract

Reviewer comments:

Add in background detail about why enrichment is used for farrowing sows.

Response: Related background detail was in line 56-66. Research focusing on the effects of housing on maternal behavior suggested that an enriched farrowing pen environment increases the possibility that sows freely make contact with their piglets, express exploratory behavior, and are able to avoid crushing their piglets [10]. A supply of nesting materials, which is provided prepartum in farrowing houses (e.g., sawdust and straw), can increase the duration and frequency of nest-building behavior prior to farrowing. This has been shown to be beneficial to parturition and the expression of maternal behavior in early-lactating sows [11-13]. Sows placed in loose farrowing environments showed a longer duration of postpartum lateral lying, which increases piglets’ access to the sows’ udders and thus piglet suckling and survival [14]. In contrast, it has been reported that a farrowing pen with straw exerts no effect on suckling duration and frequency [15, 16]. Overall, controversy persists regarding the effects of enriched environments (EE) on sow maternal behavior.

Reviewer comments:

Landrace x Yorkshire is incorrectly abbreviated

Response:

Thank you for your correction, I've changed TM into LY.

Reviewer comments:

Lines 41/42: I don’t think this claim can be substantiated by the results of the study. Inheritance of traits was not studied.

Response: Thanks for the advice, I have rewritten the conclusion as following: In conclusion, both EE increased the expression of hormones related to parental behaviors and prenatal nesting and nursing behavior of sows.

Introduction

Reviewer comments:

50: Does this refer to piglet or sow mortality?

Response:

Thank you for your notice, this refer to piglet mortality.

Reviewer comments:

51: Remove “thus”

Response:

“thus” has been removed.

Reviewer comments:

75/76: This sentence is not clear. Breed is not mentioned when citing the previous two studies, did they study breed?

Response:

Thank you for your advices. The breed of sows was Large White × Landrace and they didn’t study breed.

Reviewer comments:

80: Consider rewording “outstanding”

Response:

Thanks for your advice, I've corrected “outstanding” to “favourable”.

Reviewer comments:

130 – 140: It would be helpful here to detail the total number of hours of footage which were observed and the sampling schedule. Was footage watched continuously or in sampling intervals? From summing the hours stated, it seems that 3600 hours of footage were watched, is this correct?

Response:

Sow behavior was recorded continuously and the following footage was kept for analysis: 18 h prior to farrowing (nesting behavior), during farrowing (farrowing duration and piglet intervals), after the delivery of lucerne (enrichment use) on −2, +5 and +13 days post-farrowing, and during feeding (anticipatory behaviour) on −2, +3 and +12 days post-farrowing.

Reviewer Comments:

142: How was it known that the sow would farrow the next day? Was blood taken every day and retrospectively analysed when farrowing date was know?

Response:

We make predictions based on multiple factors including due date, clinical diagnosis and breeding experience, which can be accurate up to 1-2 days before farrowing, and blood samples were collected every day and retrospectively.

Reviewer Comments:

150: Which behaviours were state and which were events. Maybe this could be defined and added to the ethogram.

Response: Thanks for the advices.

State: Posture

Events: Nursing behavior; Nest-building behavior; Posture change

Reviewer Comments:

168 (ethogram):

Nest building duration – where does the time frame of 5 and 30s come from?  What happened if the sow spent longer than 30s building a nest?

Response: This definition comes from the literature:

Yun, J., Swan, K. M., Farmer, C., Oliviero, C., Peltoniemi, O., & Valros, A. (2014). Prepartum nest-building has an impact on postpartum nursing performance and maternal behaviour in early lactating sows. Applied animal behaviour science, 160, 31-37.

Reviewer Comments:

Nest building frequency – please add the pre-determined time period in here

Response: The pre-determined time period was the video observe time.

Reviewer Comments:

Suggestion (personal preference): Consider left aligning the table to make it easier to read.

Response: Thanks for your advice, I am already left aligning the table.

Results

Reviewer Comments:

Table 2: Single duration of nesting is not defined in the methods, please add this in to make the results easier to understand.

Response: Start of nest-building: Actively performing nest-building for longer than 5 s

End of nest-building: Not performing nest-building for longer than 30 s

(Jinhyeon et al., 2014)

Reviewer Comments:

Figure 2: Please add the unit of measurement on the y axes.

Response: Thank you for your suggestion, I have added the unit of measurement(s) on the y axes.

Reviewer Comments:

195-211: This section feels repetitive since all of the results are reported in the table. Perhaps instead the authors could summarise the key findings here and direct the reader to the table for full results.

Response: Thank you for your suggest, I've summarized the key findings and rewrote the following paragraph:

Lateral recumbency during the first three days postpartum followed a gradually decreasing tendency and differed between the observation days (p < 0.0001; Table 3). The ventral recumbency of sows on the 2nd and 3rd days postpartum was higher than at the first day postpartum (p < 0.0001). Sows in EE spent a shorter time in ventral recumbency compared with sows in BE (p < 0.05). Neither the types of environments nor the observation times showed any effect on sitting and standing (p > 0.05). LY sows spent more time sitting (p < 0.01) and less time standing (p < 0.0001) during the first three days postpartum compared with DM and LM sows. During the first three days, EE sows had a higher frequency of postural change from lateral recumbency to other postures compared with BE sows (p < 0.05). The changes among the frequency of postural changes from standing to ventral recumbency were significant between crossbreeds (DM > LM > LY; p < 0.0001).

Reviewer Comments:

Table 3: Were percentages/proportions of behaviours used in the model? Percentages are bounded by 0 and 100 and as such are not suitable for the type of model used. Beta regressions may be a useful model or poisson count models for frequencies.

Response: Thank you very much for your advice. When analyzing data, percentages are bounded by 0 and 1, and other articles have also used the GLM model (Edwards L E, Plush K J, Ralph C R, et al. Enrichment with Lucerne Hay Improves Sow Maternal Behaviour and Improves Piglet Survival[J]. Animals, 2019, 9(8): 558.). So, may I not reanalyze the data?

Reviewer Comments:

222: I think this p-values may be incorrect. I cannot see it in the table and it does not seem to match the result stated.

Response: I'm sorry I wrote this P value wrong and it has now been corrected.

Reviewer Comments:

224: Where does this p-value come from?

Response: This is the number that I've gotten rid of, But I forgot to delete it in the result, thanks for reminding.

Discussion

A re-analysis of results may well change the outcome of the discussion. I will be happy to provide a review of the discussion after the re-analysis of the data.

Submission Date

17 October 2019

Date of this review

14 Nov 2019 16:56:08

Reviewer 3 Report

Han et al., “Equipping farrowing pens with straw…”

This study investigate influence of straw provision on multiparous sows of three different breeds in farrowing pens. The response indicators was sow nest-building and postpartum maternal behaviour and concentration of oxytocin, prolactin and cortisol. The paper is generally well written, and interesting. However, I believe the authors could improve the manuscript, e.g. making several points clearer. Among other things, I need a consideration in the discussion on the use of multiparous sows.

Because: There is a selection of successful sows from first to later parities, thus by using later parities, studies include a bias (studying the most robust individuals). The effects of EE and hormones may be even larger in first parity sows? What about the breed effects, we cannot exclude that the difference between breeds are due a different selection from first to lager parties. Thus, you should consider that supplemental studies are needed on first-parity sows. You may well expect that the variations is larger, thus more than 12 replications per experimental groups may be needed (e.g. N= 20, then with straw provision only, comparing the breed of first parity sows).

Also: the previous experience in sows. Have they all farrowed with or without straw in the same system previously? This should be stated and discussed clearly, whether this could explain some of the findings.

Three other important points to be addressed before suitable for publication are:

(1) Did you use any farrowing help in this study? E.g., did you allow prostaglandin or oxytocin injection to sows (farrowing induction)? These procedures should be stated clearly as may influence the hormone and behavioural observations. If farrowing assistance (pharmacological and otherwise) then the affected number of sows per breed/treatment group must be reported. Should discuss whether these procedures may influence the findings.’

(2) For the sow postpartum behaviour – did you take the number of piglets into account? Can the number of different piglets per individual sows explain the difference between treatment (plus/minus straw) and between breeds in postpartum behaviour? The number of piglets per litter should be reported and the issue addressed in the discussion. Also report whether levelling out of litters/any cross-fostering were used.

(3) Several places, there is no mentioning of any significant interactions (e.g. Breed*E, or Breed*time or E*time, or Breed*E*time?). Thus they were none or? Please specify this more clearly in the results section.

Please find a range of other comments below:

In simple summary: please specify that both groups of sows were in farrowing crates, or in loose-housed pens?

Abstract: “ In conclusion, an EE may induce the expression of physiological hormones related to parental behaviors and thus stimulate the prenatal nesting and nursing behavior of sows.”

How can you know whether this is the causality vs. correlated responses? Could it even be that nesting and nursing behaviour gave rise to the hormonal increase in OXT and PRL? I suggest reformulating, both EE increased the expression of hormones related to parental behaviours and prenatal nesting and nursing behaviour of sows.

Line 47 Maternal crushing has been reported as the main reason for piglet mortality [2, 3]

Yes, but probably not the main cause, rather a secondary outcome due to cold, weak piglets not avoiding sow laying down? Rather causes of early piglet mortality include reduced vitality due parturition problems, hypothermia and lack of adequate colostrum intake. Besides, low vitality piglets may run a larger risk of being crushed by the sow. Please consider to specify this further in the discussion line 318ff. There has been too much weight on crushing as primary cause of piglet deaths! In the present study, I also miss a discussion of whether straw also provides improved microclimate for the piglets.

L64-76 or in the discussion: Ox and COR around farrowing also measured LY sows in the study of Malmkvist et al., 2009J. Anim. Sci. 2009. 87:2796–2805. The observed oxytocin increase and peak coincide in this study with the passive phase where sows lie laterally with less overall activity – relevant for the present study. Further, the plasma oxytocin concentration (measured hourly) increased during farrowing, and maximum concentrations were reached late in the farrowing rather than in the beginning. Later after birth: more variation due to lactation-induced secretion. Thus, I miss a time consideration in the statistic treatment – because if you take the blood sample at a fixed time every day, then the distance to farrowing (birth of first piglet) varies. Have you considered this in the statistical analysis? Can different times of farrowing explain your findings of different oxytocin concentrations?

L81 Please provide this average number with a variation (s.d.?), is it given per delivering or per pregnant sow?

L84-85. Consider to state the hypothesis and predictions more clearly. EE (straw provision) induce more maternal behaviour and less cortisol in sows around parturition? LY less maternal behaviour? Barren: more cortisol, less OX and PRL? What about litter size- you would expect that a larger number of piglets increase OX during the lactation period – or did you have the exact same litter size? You did not include litter size in the models. Please also include the piglet effect in the discussion and interpretation of the results.

L94 or L107. Please specify the type and length of straw (e.g. Barley?). Given on the floor is better than in a hay rack (several studies), maybe cite them?

L100. The sows were probably of a different size, but the same size of pen were used. Thus, had the sows equal opportunity to perform behaviours, or was e.g. the large LY sows more restricted? Also L300 you discuss the size of the pen – so please discus this.

L142 Oxytocin and cortisol concentration change relative to the time of farrowing (cf. Malmkvist et al. 2009). Did you that duration (time) to farrowing into account in the statistical analysis, on the farrowing day? What if sows have had farrowed before 10:00 – is this sample counted as post-farrowing? These considerations should be dealt with in the manuscript.

L150 what is meant by ‘preliminarily’ here? Some calculations made in excel. Please specify.

L176 Table 2. I suggest to help the readers more for this table, and to minimise the foot notes. Specifically, ‘behavior’ in the heading could be ‘Prepartum nest-building behavior’. FN, DN and SN should be written out, frequency, total duration and bout duration.

I also suggest that you keep one column for housing condition, written out (Barren, Enriched), and reserve one column for the breeds. Only for the breeds, abbreviations are fine.

In the title of the Table, please specify when is the prepartum period in days or hours. AND for how long each sow were observed. Reduce the number of decimals in the table 2. I think that 0 would be sufficient for frequencies, e.g. 39 ±16.5 instead of 38.83  ±16.46. Also since you probably only counted whole numbers. Duration minutes: 45.7 ±33.57 would be fine. For bout length, you give the values down to 0.01 seconds – but I doubt that you really look that precisely on the behaviour using Noldus. Often the minimum observation precision is something like 1 seconds. Thus, I suggest keeping the report to whole seconds, as to no overestimate the actual observational precision. For example 72.27 ±98.17 seconds become 72 ±98.2 s. This will make the table easier to read.

Delete not necessary info in the footnote (write it properly in the table instead). However, you need to explain the different lettering A, B and a, b, not explained at all.

When you give the Enrichment and Breed p-values separately, this is because there was no significant ExB interaction? Please specify the ExB p-value. I think you should write the test statistics as well (Fn, n = x.x, p = 0.xxx).

Throughout the results section only p-values are given. However, it is mandatory to give the  test statistics and DF. This helps to see whether the statistics can evaluated as OK. For example if too high DF, this can be due to pseudoreplication, not using correct code for analysing repeated measures. Please add the needed test statistics and DF, before the paper can be considered for publication.

Figure 2. Please add the x-axis time scale (hours). I guess there is a Breed x Environment interaction? The x-axis could be pre-farrowing time (hours). Please use the same scale for all three y-axis (now 1200, 1600 and 1600 as max. values). Then the breeds are more comparable. One y-axis title on the left could be enough – omit the others to make it easier to read them. I suggest that the labels only are Barren and Enriched (rather than LYB, LYE etc.), as you have the breed in the headline of each figure. Then maybe writhe the breeds out.

Also in table 3-5: to many decimal points. E.g. for non-significant P = 0.35 is enough. For P < 0.001 this is enough to write P < 0.001, and P = 0.078 is good. Is there no interactions between Environment and Breed? Or breed*environment*day?

L258 please change title Physiological indexes to something more meaningful. E.g. Hormone concentrations. Not any interactions? For the day before farrowing and the day of farrowing, you need to take the exact birth time into account – could be used in the models. Because the concentration of hormones may be influenced by the farrowing. There is no mentioning of any significant interactions? (e.g. Breed*E, or Breed*time or E*time, or Breed*E*time?).

For the sow postpartum behaviour – did you take the number of piglets into account? Can the number of different piglets per individual sows explain the difference between treatment (plus/minus straw) and between breeds in postpartum behaviour? This should be addressed.

L351 please specify whether the litter size was equal between the Breed and E treatment – may a different number of piglets explain the postpartum results?

L391 I miss a discussion of Cortisol concentration in relation to farrowing. That high Cortisol may correlate with farrowing that are more problematic? Or how did you deal with this.

Author Response

Reviewer 3

©©Open Review

English language and style

( ) Extensive editing of English language and style required
( ) Moderate English changes required
(x) English language and style are fine/minor spell check required
( ) I don't feel qualified to judge about the English language and style

Yes

Can be improved

Must be improved

Not applicable

Does the introduction provide sufficient background and include all relevant references?

(x)

( )

( )

( )

Is the research design appropriate?

( )

(x)

( )

( )

Are the methods adequately described?

( )

( )

(x)

( )

Are the results clearly presented?

( )

( )

(x)

( )

Are the conclusions supported by the results?

(x)

( )

( )

( )

Comments and Suggestions for Authors

Han et al., “Equipping farrowing pens with straw…”

This study investigate influence of straw provision on multiparous sows of three different breeds in farrowing pens. The response indicators was sow nest-building and postpartum maternal behaviour and concentration of oxytocin, prolactin and cortisol. The paper is generally well written, and interesting. However, I believe the authors could improve the manuscript, e.g. making several points clearer. Among other things, I need a consideration in the discussion on the use of multiparous sows.

Reviewer comments:

Because: There is a selection of successful sows from first to later parities, thus by using later parities, studies include a bias (studying the most robust individuals). The effects of EE and hormones may be even larger in first parity sows? What about the breed effects, we cannot exclude that the difference between breeds are due a different selection from first to lager parties. Thus, you should consider that supplemental studies are needed on first-parity sows. You may well expect that the variations is larger, thus more than 12 replications per experimental groups may be needed (e.g. N= 20, then with straw provision only, comparing the breed of first parity sows).

Response: Thank you for your intensive reading and pertinent advice. First, all sows are selected through strict breeding and thus have similar body conditions and reproductive performance. Second, this paper focuses on maternal behavior, while first parity sow’s performance weaker maternal than later parities, hence I have chosen three to five parity sows as research object. Third, to exclude the difference between parity, the number of sows was equality among three to five parity sows.

Reviewer comments:

Also: the previous experience in sows. Have they all farrowed with or without straw in the same system previously? This should be stated and discussed clearly, whether this could explain some of the findings.

Response: Thank you for the reminder. The previous experience in sows, they all farrowed without straw in the same system previously, which is also the premise of experimental unified treatment.

Three other important points to be addressed before suitable for publication are:

Reviewer comments:

(1) Did you use any farrowing help in this study? E.g., did you allow prostaglandin or oxytocin injection to sows (farrowing induction)? These procedures should be stated clearly as may influence the hormone and behavioural observations. If farrowing assistance (pharmacological and otherwise) then the affected number of sows per breed/treatment group must be reported. Should discuss whether these procedures may influence the findings.’                                                                                                                                                                                                                                                                                                                                                                                                                                                                                                                                                                                                                                                                                                                                                                                                                                                                                  

Response: Thank you for your suggestion. I didn’t use any farrowing help in this study, all sows were farrowing naturally.

Reviewer comments:

(2) For the sow postpartum behaviour – did you take the number of piglets into account? Can the number of different piglets per individual sows explain the difference between treatment (plus/minus straw) and between breeds in postpartum behaviour? The number of piglets per litter should be reported and the issue addressed in the discussion. Also report whether levelling out of litters/any cross-fostering were used.

Response: Our sows are strictly bred, moreover, the reproductive performance of sows from three to five parity was relatively stable. The litter size of each group of sows was between 13 and 15: 13.33±2.73; 13.33±1.86; 14.00±2.19; 13.33±1.86;  15.33±2.50; 13.50±2.42, so I think the number of different piglets per individual sows maybe couldn’t lead the difference between treatment. Finally, I try to eliminate external interference during the experiment so that levelling out of litters/any cross-fostering were didn’t used.

Reviewer comments:

(3) Several places, there is no mentioning of any significant interactions (e.g. Breed*E, or Breed*time or E*time, or Breed*E*time?). Thus they were none or? Please specify this more clearly in the results section.

Response: There is no mentioning of any significant interactions several places, for all values were greater than 0.05. In general, there was no significant difference between them so that I think there is no point in discussing them.

Please find a range of other comments below:

Reviewer comments:

In simple summary: please specify that both groups of sows were in farrowing crates, or in loose-housed pens?

Response: Thank you for your suggestions, both groups of sows were in loose-housed pens and I have added this sentence to the manuscript (line 14-15).

Reviewer comments:

Abstract: “In conclusion, an EE may induce the expression of physiological hormones related to parental behaviors and thus stimulate the prenatal nesting and nursing behavior of sows.”

How can you know whether this is the causality vs. correlated responses? Could it even be that nesting and nursing behaviour gave rise to the hormonal increase in OXT and PRL? I suggest reformulating, both EE increased the expression of hormones related to parental behaviours and prenatal nesting and nursing behaviour of sows.

Response: I think what your suggestion makes a lot of sense, I have been too hasty in drawing this conclusion. I have made the following changes to the conclusion: both EE increased the expression of hormones related to parental behaviours and prenatal nesting and nursing behaviour of sows.

Reviewer comments:

Line 47 Maternal crushing has been reported as the main reason for piglet mortality [2, 3]

Yes, but probably not the main cause, rather a secondary outcome due to cold, weak piglets not avoiding sow laying down? Rather causes of early piglet mortality include reduced vitality due parturition problems, hypothermia and lack of adequate colostrum intake. Besides, low vitality piglets may run a larger risk of being crushed by the sow. Please consider to specify this further in the discussion line 318ff. There has been too much weight on crushing as primary cause of piglet deaths! In the present study, I also miss a discussion of whether straw also provides improved microclimate for the piglets.

 Response: After the piglets were born, piglets rest region will install the insulation board and the insulation lamp to heating, so I don't think there's a problem with cold in piglets. Besides, low vitality piglets may run a larger risk of being crushed by the sow, but it's out of control and the number of weak piglets is relatively small, so I didn't talk about this situation. Maybe straw also provides improved microclimate for the piglets (Tuyttens, 2005), I already added it to the manuscript.

Reviewer comments:

L64-76 or in the discussion: Ox and COR around farrowing also measured LY sows in the study of Malmkvist et al., 2009J. Anim. Sci. 2009. 87:2796–2805. The observed oxytocin increase and peak coincide in this study with the passive phase where sows lie laterally with less overall activity – relevant for the present study. Further, the plasma oxytocin concentration (measured hourly) increased during farrowing, and maximum concentrations were reached late in the farrowing rather than in the beginning. Later after birth: more variation due to lactation-induced secretion. Thus, I miss a time consideration in the statistic treatment – because if you take the blood sample at a fixed time every day, then the distance to farrowing (birth of first piglet) varies. Have you considered this in the statistical analysis? Can different times of farrowing explain your findings of different oxytocin concentrations?

Response: Your suggestion is reasonable, but we mainly analyze the level of oxytocin prior to, in and after farrowing each sow. If consistent interval, we will affect farrow time and so on, it will interfere with the delivery time and delivery interval. Therefore, we adopt the fixed time collection. What we want to express is the change of oxytocin during farrowing.

Reviewer comments:

L81 Please provide this average number with a variation (s.d.?), is it given per delivering or per pregnant sow?

Response: I'm sorry it wasn't in the original article so that I can’t provide this average number with a variation. It is given per delivering of each sow.

Reviewer comments:

L84-85. Consider to state the hypothesis and predictions more clearly. EE (straw provision) induce more maternal behaviour and less cortisol in sows around parturition? LY less maternal behaviour? Barren: more cortisol, less OX and PRL? What about litter size- you would expect that a larger number of piglets increase OX during the lactation period – or did you have the exact same litter size? You did not include litter size in the models. Please also include the piglet effect in the discussion and interpretation of the results.

Response:

Reviewer comments:

L94 or L107. Please specify the type and length of straw (e.g. Barley?). Given on the floor is better than in a hay rack (several studies), maybe cite them?

Response: In this experiment, dry rice stalks were added, the length of straw about 50cm and I gave it on the floor.

Reviewer comments:

L100. The sows were probably of a different size, but the same size of pen were used. Thus, had the sows equal opportunity to perform behaviours, or was e.g. the large LY sows more restricted? Also L300 you discuss the size of the pen – so please discus this.

Response: The size of pens is large enough for sows to express all behaviors. Therefore, the LY sows didn’t be restricted.

Reviewer comments:

L142 Oxytocin and cortisol concentration change relative to the time of farrowing (cf. Malmkvist et al. 2009). Did you that duration (time) to farrowing into account in the statistical analysis, on the farrowing day? What if sows have had farrowed before 10:00 – is this sample counted as post-farrowing? These considerations should be dealt with in the manuscript.

Response: Indeed, oxytocin and cortisol concentration change relative to the time of farrowing. Maybe we didn't pay attention to that detail.

Reviewer comments:

L150 what is meant by ‘preliminarily’ here? Some calculations made in excel. Please specify.

Response: We organized the raw data in excel and figure out the behavior time ratio.

Reviewer comments:

L176 Table 2. I suggest to help the readers more for this table, and to minimise the foot notes. Specifically, ‘behavior’ in the heading could be ‘Prepartum nest-building behavior’. FN, DN and SN should be written out, frequency, total duration and narrow width.

I also suggest that you keep one column for housing condition, written out (Barren, Enriched), and reserve one column for the breeds. Only for the breeds, abbreviations are fine.

Response: Thanks for your advices. However, considering too much data, the font should not be too small, so we choose abbreviations to narrow width. If I write the full name, then I need to break up the table into several, which will lead the table fuzzy.

Reviewer comments:

In the title of the Table, please specify when is the prepartum period in days or hours. AND for how long each sow were observed. Reduce the number of decimals in the table 2. I think that 0 would be sufficient for frequencies, e.g. 39 ±16.5 instead of 38.83  ±16.46. Also since you probably only counted whole numbers. Duration minutes: 45.7 ±33.57 would be fine. For bout length, you give the values down to 0.01 seconds – but I doubt that you really look that precisely on the behaviour using Noldus. Often the minimum observation precision is something like 1 seconds. Thus, I suggest keeping the report to whole seconds, as to no overestimate the actual observational precision. For example 72.27 ±98.17 seconds become 72 ±98.2 s. This will make the table easier to read.

Response: Since I calculated the average of multiple Numbers, the decimal point appears. But I think your suggestion is very pertinent, at the same time to ensure the accuracy of the data, so I'll keep it to one decimal place.

Delete not necessary info in the footnote (write it properly in the table instead). However, you need to explain the different lettering A, B and a, b, not explained at all.

Reviewer comments:

When you give the Enrichment and Breed p-values separately, this is because there was no significant ExB interaction? Please specify the ExB p-value. I think you should write the test statistics as well (Fn, n = x.x, p = 0.xxx).

Response: Thank you for your advice, there was no significant ExB interaction so I didn’t write it out. I think there's few points in talking about that so there's no writing. If I annotate every paragraph, will the article be too tedious?

Reviewer comments:

Throughout the results section only p-values are given. However, it is mandatory to give the test statistics and DF. This helps to see whether the statistics can evaluated as OK. For example if too high DF, this can be due to pseudoreplication, not using correct code for analysing repeated measures. Please add the needed test statistics and DF, before the paper can be considered for publication.

Response: I'm sorry I didn't catch your meaning. How can I give the give the test statistics and DF? Add some columns to the back of the form?

Reviewer comments:

Figure 2. Please add the x-axis time scale (hours). I guess there is a Breed x Environment interaction? The x-axis could be pre-farrowing time (hours). Please use the same scale for all three y-axis (now 1200, 1600 and 1600 as max. values). Then the breeds are more comparable. One y-axis title on the left could be enough – omit the others to make it easier to read them. I suggest that the labels only are Barren and Enriched (rather than LYB, LYE etc.), as you have the breed in the headline of each figure. Then maybe writhe the breeds out.

Response: Thanks for the advices. I have added the x-axis time scale (hours). Since the video viewing time is accurate to seconds, to make the data more accurate and the trend more obvious.

Reviewer comments:

Also in table 3-5: to many decimal points. E.g. for non-significant P = 0.35 is enough. For P < 0.001 this is enough to write P < 0.001, and P = 0.078 is good. Is there no interactions between Environment and Breed? Or breed*environment*day?

Response: Thank you very much, I have made the correction all tables according to your request.

Reviewer comments:

L258 please change title Physiological indexes to something more meaningful. E.g. Hormone concentrations. Not any interactions? For the da act birth time into account – could be used in the models. Because the concentration of hormones may be influenced by the farrowing. There is no mentioning of any significant interactions? (e.g. Breed*E, or Breed*time or E*time, or Breed*E*time?).y before farrowing and the day of farrowing, you need to take the ex

Response: Thank you very much, I have change Physiological to stress.

For the sow postpartum behaviour – did you take the number of piglets into account? Can the number of different piglets per individual sows explain the difference between treatment (plus/minus straw) and between breeds in postpartum behaviour? This should be addressed.

Response: Our sows are strictly bred, moreover, the reproductive performance of sows from three to five parity was relatively stable. The litter size of each group of sows was between 13 and 15: 13.33±2.73; 13.33±1.86; 14.00±2.19; 13.33±1.86;  15.33±2.50; 13.50±2.42, so I think the number of different piglets per individual sows maybe couldn’t lead the difference between treatment. Finally, I try to eliminate external interference during the experiment so that levelling out of litters/any cross-fostering were didn’t used.

Reviewer comments:

L351 please specify whether the litter size was equal between the Breed and E treatment – may a different number of piglets explain the postpartum results?

Response: Results for piglets are as follows:

LYB

LYE

DMB

DME

LMB

LME

E

B

D

Litter size

13.33±2.73

13.33±1.86

14.00±2.19

13.33±1.86

15.33±2.50

13.50±2.42

0.2829

0.5013

0.615

Both breed and E treatment have no significant impact on litter size.

Reviewer comments:

L391 I miss a discussion of Cortisol concentration in relation to farrowing. That high Cortisol may correlate with farrowing that are more problematic? Or how did you deal with this.

 Response: The concentration of cortisol is an indicator of stress. However, when sow farrowing, the stress of sows was too much so that

 data have no reference.

Submission Date

17 October 2019

Date of this review

17 Nov 2019 15:57:59

Round 2

Reviewer 2 Report

Equipping farrowing pens with straw improves maternal behavior and physiology of Min-pig hybrid sows: Review 2

The authors have not addressed any of my main points with the paper. In my general comments (re-pasted below), I raised concerns about the statistics. The results regarding behaviour are gained by comparing two sets of animals which have different behavioural opportunities. Sows without nesting material cannot perform the nesting behaviours outlined in the ethogram (aside from rooting and pawing) and so a difference is bound to be seen when comparing to sows with nesting material. Therefore the analysis is not appropriate for this study. I provided an alternative option, which the authors did not address, nor did they provide evidence or arguments for analysing in their selected way, which I would be happy to consider.

I also asked if the authors could provide additional test statistics to P-values, as P-values alone are not suitable for interpreting results (see: https://www.nature.com/articles/d41586-019-00857-9), but this has not been addressed.

Response to detailed comments from Review Round 1:

I am still unclear on the total hours of footage observed. Could the authors please provide the total after the explanation? Please clarify what happened with behavioural coding if the sow spent longer than 30s building a nest as this is not clear from the ethogram or text.

Original comments from Review Round 1:

This work is important and adds to the current literature in this area, but in my opinion, needs re-analysis to draw the conclusions the authors currently state. Analysing nesting as an aggregation of all nesting behaviours means that significant results are bound to be seen, as enriched pigs have the opportunity to build nests and barren pigs do not. A stronger and more useful analysis would be to separate out manipulation of nesting behaviours (and analyse the effect of breed on these) from non-manipulation behaviours (rooting and pawing) which can be analysed for breed and enrichment effects. Parity is currently not included in the model, but could be an important factor of maternal behaviour. Time should be included in both models if the authors wish to explore effects of time on pre-partum behaviour (lines 183-191). Throughout the results, the interactions are not reported, though they are included in the methods. Please could the authors make it clear whether interactions were significant and if not, whether the interaction was removed before a model re-run? I would suggest doing this with the re-analysis. In addition, effect measures are not reported with only p-values throughout the text.

Throughout the paper uses abbreviations which are confusing and make the manuscript difficult to follow. I suggest replacing these abbreviations with the full terms for the physiological measures and behaviours (crossbreed abbreviations could be kept).

Author Response

Reviewer comments:

The authors have not addressed any of my main points with the paper. In my general comments (re-pasted below), I raised concerns about the statistics. The results regarding behaviour are gained by comparing two sets of animals which have different behavioural opportunities. Sows without nesting material cannot perform the nesting behaviours outlined in the ethogram (aside from rooting and pawing) and so a difference is bound to be seen when comparing to sows with nesting material. Therefore the analysis is not appropriate for this study. I provided an alternative option, which the authors did not address, nor did they provide evidence or arguments for analysing in their selected way, which I would be happy to consider.

 Respond: Thanks for your suggestion which is very pertinent. Nevertheless, please give me an explanation of the reason. First, although the performance of nesting behaviour can vary, its occurrence is innately typical and relatively unnaffected by environmental and social constraints (Arey D S. Straw and food as reinforcers for prepartal sows[J]. Applied Animal Behaviour Science, 1992, 33(2-3): 217-226.). Second, there are many GLM procedures for similar data, as follows:

Edwards L E, Plush K J, Ralph C R, et al. Enrichment with Lucerne Hay Improves Sow Maternal Behaviour and Improves Piglet Survival[J]. Animals, 2019, 9(8): 558. Liu H, Yi R, Wang C, et al. Behavior and physiology of two different sow breeds in a farrowing environment during late 35-day lactation[J]. PloS one, 2018, 13(5): e0197152.

In general, I think maybe GLM procedure can be used to analyze the data in this manuscript.

Reviewer comments:

I also asked if the authors could provide additional test statistics to P-values, as P-values alone are not suitable for interpreting results (see: https://www.nature.com/articles/d41586-019-00857-9), but this has not been addressed.

Respond: Thank you very much for your suggestion. I added the F value as additional test statistics.

Reviewer comments:

Response to detailed comments from Review Round 1:

I am still unclear on the total hours of footage observed. Could the authors please provide the total after the explanation? Please clarify what happened with behavioural coding if the sow spent longer than 30s building a nest as this is not clear from the ethogram or text. 

Respond: The total hours of footage observed was 18h. Maybe I don't have a clear definition of nesting behavior.

Start of nest-building: Actively performing nest-building for longer than 5 s.

End of nest-building: Not performing nest-building for longer than 30 s.

The time in between is counted in the nesting duration.

 Reviewer comments:

Original comments from Review Round 1:

This work is important and adds to the current literature in this area, but in my opinion, needs re-analysis to draw the conclusions the authors currently state. Analysing nesting as an aggregation of all nesting behaviours means that significant results are bound to be seen, as enriched pigs have the opportunity to build nests and barren pigs do not. A stronger and more useful analysis would be to separate out manipulation of nesting behaviours (and analyse the effect of breed on these) from non-manipulation behaviours (rooting and pawing) which can be analysed for breed and enrichment effects.

Respond: Thank you for your advice. Although the performance of nesting behaviour can vary, its occurrence is innately typical and relatively unnaffected by environmental and social constraints (Arey D S. Straw and food as reinforcers for prepartal sows[J]. Applied Animal Behaviour Science, 1992, 33(2-3): 217-226.). Hence, in my opinion, maybe separate out manipulation of nesting behaviours (and analyse the effect of breed on these) from non-manipulation behaviours (rooting and pawing) will broke the definition of nesting behavior.

 Reviewer comments:

Parity is currently not included in the model, but could be an important factor of maternal behaviour. Time should be included in both models if the authors wish to explore effects of time on pre-partum behaviour (lines 183-191).

Respond: Indeed, parity could be an important factor of maternal behavior. However, the subject of this article is environment and crossbreed. At the same time, we have the same number of sows for each parity in each group, so we have excluded the effect of parity.

 Reviewer comments:

Throughout the results, the interactions are not reported, though they are included in the methods. Please could the authors make it clear whether interactions were significant and if not, whether the interaction was removed before a model re-run? I would suggest doing this with the re-analysis. In addition, effect measures are not reported with only p-values throughout the text.

Respond: I think you have the right idea about interaction. Some indicators have not interacted and have been reanalyzed.

 Reviewer comments:

Throughout the paper uses abbreviations which are confusing and make the manuscript difficult to follow. I suggest replacing these abbreviations with the full terms for the physiological measures and behaviours (crossbreed abbreviations could be kept).

Respond: Thank you for your advice. I have changed all abbreviations to full terms.
